# Naked-Eye Chromogenic Test Strip for Cyanide Sensing Based on Novel Phenothiazine Push–Pull Derivatives

**DOI:** 10.3390/bios12060407

**Published:** 2022-06-13

**Authors:** Pedro E. Martín Várguez, Jean-Manuel Raimundo

**Affiliations:** Aix Marseille Univ, CNRS, CINaM, 13005 Marseille, France; pedro.martinva@uanl.edu.mx

**Keywords:** push–pull, chromophore, phenothiazine, cyanide, acetate, optical detection

## Abstract

Monitoring and detection of cyanide are of crucial interest as the latter plays versatile roles in many biological events, is ubiquitous in environment, and responsible for several acute poisoning and adverse health effects if ingested. We describe herein the synthesis and characterization of novel phenothiazine-based push–pull chromogenic chemosensors suitable for naked eye cyanide sensing. Indeed, specific detections were achieved for cyanide with a LOD of *ca* 9.12 to 4.59 µM and, interestingly, one of the new chemosensors has also revealed an unprecedented affinity for acetate with a LOD of *ca* 2.68 µM. Moreover, as proof of concept for practical applications, a paper test strip was prepared allowing its use for efficient qualitative naked eye cyanide sensing.

## 1. Introduction

Anionic species are a major class of structures involved in many physiological, biological, chemical, and environmental events and processes [1]. Owing to their central roles, the design and synthesis both of artificial and bioinspired probes of anion is of crucial interest for their detection in real samples [2]. Among them, cyanide constitutes one of the most toxic anionic species because it is able to readily bind hemoglobin’s iron and interferes with many vital processes [3,4,5]. Nevertheless, despite its harmful effects, it is still widely used in mining or metallurgical industries and in plastic production plants. Thus, industrial accidents or ineffective effluent treatments are the main sources of large cyanide spills into water and environment. According to the WHO, cyanide concentration should not exceed 1.9 µM in drinking water [6]. In addition, accidental ingestion from the cyanogenesis of cyanogenic glycosides present in stone fruit kernels (bitter almonds, apricot), cassava, sorghum, linseed, etc. is also responsible of acute poisoning [7] requiring the development of efficient and reliable sensing tools.

During the last decades, important efforts have been devoted to developing efficient analytical techniques such as ion chromatography coupled with pulse amperometric detection [8] or capillary electrophoresis [9] but they require sophisticated equipment, expert manpower, and are time-consuming. In addition, a preconcentration or a filtering step is usually needed, acting as a drag in real-time detection or analysis at the point-of-need. To overcome these issues, scientists have strived to design and develop optical probes for fast cyanide sensing [2,10]. In peculiar, naked eye techniques based on colorimetric sensors are more suitable for in situ sensing and have drawn deep attention due to their low-cost instrumentation, ease-of-use, nondestructive determination, specificity, and selectivity.

To this aim, supramolecular chemistry based on noncovalent interactions has aroused particular interest in anion sensing [11]. Indeed, a host–guest recognition induces some photophysical changes associated with a detectable optical output signal. Based on their structures and mode of action, those chromogenic/fluorogenic chemosensors are categorized in different categories, namely, two- or three-component, multicomponent, and inorganic–organic hybrid, which have been exploited and used in cyanide sensing [10,12,13,14]. Furthermore, putative competitive ions remain one of the main problems to be solved in order to improve the specificity and selectivity toward the desired analyte. For this purpose, the nucleophilic properties of cyanide have been exploited in order to avoid or limit the competition of interferent ions. Those include the nucleophilic substitution reactions of the cyanide with a variety of electron poor chemical structures such as squaraine [15], acridinium [16], pyrylium [17], indolium [18], oxazine [19], trifluoroacetamide or trifluoroacetophenone derivatives [20], dicyanovinyl electrophilic groups derivatives [21], and so on. Although, important improvements achieved in the design and synthesis of naked eye-based chemosensors is still of great interest, with the main challenge to conceive structures exhibiting high sensitivity and selectivity.

Considering its intrinsic optical properties, phenothiazine constitutes a promising candidate as signaling scaffold for the development of optical chemosensors. Phenothiazine acts as electron donor due to the presence of electron-rich nitrogen and sulfur atoms. We sought to combine the aromatic core with different electron-withdrawing groups, as recognition moiety, in order to synthesize novel phenothiazine-based push–pull (D-π-A) chemosensors that will be optically sensitive to the nucleophilic addition of the cyanide. Interestingly, few phenothiazine derivatives [22,23,24,25,26,27,28,29,30,31,32,33] have been designed and synthetized for cyanide sensing and all are based on the direct substitution of the phenothiazine scaffold with electron-poor aromatic rings. Based on these considerations, we report herein on the synthesis of a set of new push–pull chromophores, where the substitution with dicyano-based acceptors occurred at the nitrogen position linked to a benzene ring as electron relay between the donor and acceptor parts. In addition, the functionalization of the phenothiazine pattern at the nitrogen atom vanished its intrinsic fluorescence properties, making the newly synthetized compounds as only colorimetric sensors.

## 2. Materials and Methods

### 2.1. General

All reactants and solvents were analytical grade, purchased from Alfa Aesar and used as received. Acetonitrile spectrophotochemical grade was used for the optical properties and electrochemical analysis. The salt solutions of PF_6_^−^, Cl^−^, Br^−^, CN^−^, HSO_4_^−^, NO_2_^−^, and CH_3_CO_2_^−^ were prepared from their respective tetrabutylammonium derivatives. UV–visible absorption spectra were obtained on a Varian Cary 1E spectrophotometer at room temperature. Limit of detection (LOD) was determined according to the formula given by the IUPAC [34,35]. LOD_u_ = 3.3 *s_y/x_* (1 + *h*_0_ +1/*I*)^1/2^/*m*, where *s_y/x_* is the residual standard deviation, *h*_0_ is the leverage of the sample, *m* is the regression slope, and *I* is the number of samples. NMR spectra were recorded on a JEOL ECS400 NMR spectrometer at room temperature. Electrochemical studies were performed on a VersaSTAT 4 potentiostat from Princeton Applied Research (Hi Tech Detection Systems, Massy, France ). A three-electrode system based on a platinum (Pt) working electrode (ø: 1.6 mm), a Pt counter electrode, and an Ag/AgCl reference electrode was used. Bu_4_N^+^PF_6_^−^ served as a supporting electrolyte (0.1 M). All experiments were carried out in CH_3_CN at 20 °C. Electrochemical potential values versus Ag/AgCl were determined from the cyclic voltammogram at a concentration of 1.10^−3^ M with a scan rate of 100 mV·s^−1^. NMR chemical shifts are given in ppm (δ) relative to Me_4_Si with solvent resonances used as internal standards (CD_2_Cl_2_ 5.32 ppm for ^1^H and 53.5 for ^13^C; CD_3_CN: 1.93 ppm for ^1^H and 1.30; 117.7 for ^13^C; DMSO-*d*_6_: 2.50 ppm for ^1^H and 39.5 for ^13^C). The electronic absorption maxima (λ_max_) are directly extracted from absorption spectra of the tested molecules. Under the optimum conditions, the stoichiometry between the synthesized chemosensors and the different analytes were investigated by the molar ratio method by UV−visible spectroscopy. Melting points were uncorrected and obtained from a Stuart melting point apparatus SMP30. High-resolution mass spectrometry (HRMS) was performed on a SYNAPT G2 HDMS (Waters) spectrometer with an electrospray ionization source (ESI) and a TOF mass analyzer at Aix-Marseille Université Spectropole [36]. The structures and energy levels of the target compounds were calculated using the B3LYP/6-31G* level of theory and method implemented in the Gaussian 09 package. The theoretical spectrum was obtained with a Matlab script using the *lsqnonlin* function from the experimental data, stoichiometry, and molar extinction coefficient of the studied chromophores according to reference. A detailed description is given in the Appendix A.

### 2.2. Synthesis

*4-(10H-Phenothiazine-10-yl) Benzaldehyde* (**1**). In a 50-mL round flask, phenothiazine (2.440 g, 12.0 mmol), 4-bromebenzaldehyde (2.523 g, 12.3 mmol), palladium acetate (0.225 g, 1.0 mmol), tri-*tert*-butylphosphine (297 μL, 1.2 mmol), and anhydrous K_2_CO_3_ (4.188 g, 30.0 mmol) were dissolved in 20 mL of dry toluene and heated at 110 °C under an argon atmosphere for 24 h. After removing the excess of solvent, 30 mL of a 2 M aqueous HCl solution was added then extracted with 40 mL of ethyl acetate (3×) and the organic phases were dried over MgSO_4_. The obtained brownish oil was further purified by column chromatography over SiO_2_ using ethyl acetate/petroleum ether 1:4 as eluent (Rf = 0.4). Recrystallisation in pentane gives 2.840 g of a beige solid (yield 76%). ^1^H-NMR (CD_3_CN, 400 MHz). δ 9.84 (s, 1H), 7.77 (m, 2H), 7.47 (dd, *J* = 7.7, 1.3, 2H), 7.33 (dtd, *J* = 9.5, 8.1, 1.4, 4H), 7.23 (m, 2H), 7.15 (m, 2H). ^13^C-NMR (CD_3_CN, 101 MHz, ppm). δ 191.66 (-CHO), 151.02 (N-C), 142.26 (N-C), 132.68, 132.36 (Ar-C), 131.79 (Ar-C), 129.51 (Ar-C), 128.62 (S-C), 127.05 (Ar-C), 126.47 (Ar-C). Mass analysis: [M + H]^+^
*m*/*z* 304.1, [2M + 1]^+^; 607.2. Elemental analysis for C_19_H_13_ONS: Calculated, C—75.22, H—4.32, O—5.27, N—4.62, S—10.57; found, C—75.17, H—4.32, O—5.67, N—4.53, S—10.31. Melting point: 106.6 °C.

*2-(4-(10H-Phenothiazin-10-yl)benzylidene)malononitrile* (**6**). In a 25-mL round flask, 4-(10*H*-phenothiazine-10-yl) benzaldehyde **1** (303,4 mg, 1.0 mmol) and malononitrile **3** (233.6 mg, 3.5 mmol) were dissolved in 12 mL of absolute ethanol and heated at 85 °C under an argon atmosphere for 24 h. After removing the excess solvent, the solid was purified by column chromatography over SiO_2_ using CH_2_Cl_2_/C_6_H_12_ (1:1) as eluent (Rf = 0.35) affording 218.4 mg of an orange solid (yield 62%). ^1^H-NMR ((CD_3_)_2_SO, 400 MHz). δ 8.30 (s, 1H), 7.87 (d, *J* = 9.05, 2H), 7.62 (dd, *J* = 0.91, 7.71, 2H), 7.56 (dd, *J* = 0.54, 7.72, 2H), 7.47 (td, *J* = 1.17, 7.78, 2H), 7.35 (td, *J* = 0.89, 7.75, 2H), 7.11 (d, *J* = 9.04, 2H). ^13^C-NMR. ((CD_3_)_2_SO, 101 MHz, ppm). δ 159.79 (=CH), 150.01 (Ar-C), 139.82 (Ar-C), 133.25 (Ar-C), 132.85 (Ar-C), 129.05 (Ar-C), 129.04 (Ar-C), 126.90 (Ar-C), 114.96 (Ar-C), 114.22 (Ar-C), 115.15 (-CN), 75.36 (=C). Elemental analysis for C_22_H_13_N_3_S-0.05 CH_2_Cl_2_-0.1C_6_H_12_: Calculated, C—74.72, H—3.96, N—11.54, S—8.81; found, C—74.52, H—3.65, N—11.71, S—8.42. HRMS (ESI^+^): Calculated for C_22_H_13_N_3_S (*m*/*z* 351.0830), [M + H]^+^ 352.0903; found, [M + H]^+^ 352.0901. Melting point: 220.3 °C.

*(Z)-2-(2-(4-(10H-Phenothiazin-10-yl)benzylidene)-3-oxo-2,3-dihydro-1H-inden-1-ylidene)malononitrile* (**7**). In a 25-mL round flask, 4-(10H-phenothiazine-10-yl) benzaldehyde **1** (189.61 mg, 0.625 mmol) and 2-(3-oxo-2,3-dihydro-1*H*-inden-1-ylidene)malononitrile **4** (97.1 mg, 0.5 mmol) were dissolved in 4 mL of acetic anhydride and heated at 100 °C for 3 days. After removing the excess solvent, the solid was washed using a mixture of C_6_H_14_/acetone (7:1) affording 140.5 mg of a red solid (yield 58%). ^1^H-NMR. (CD_2_Cl_2_, 400 MHz) δ 8.62 (d, *J* = 7.8 Hz, 1H), 8.43 (s, 1H), 8.18 (d, *J* = 9.1 Hz, 2H), 7.85 (dd, *J* = 6.9, 1.7 Hz, 1H), 7.73 (dqd, *J* = 14.4, 7.4, 1.2 Hz, 2H), 7.51 (ddd, *J* = 7.6, 4.2, 1.1 Hz, 4H), 7.39 (td, *J* = 7.7, 1.4 Hz, 2H), 7.26 (td, *J* = 7.7, 1.2 Hz, 2H), 7.11 (d, *J* = 9.1 Hz, 2H). ^13^C-NMR. (CD_2_Cl_2_, 101 MHz, ppm) δ 187.16, 163.05, 150.90, 147.38, 140.35, 139.99, 137.78, 135.37, 134.77, 129.36, 127.84, 127.18, 126.32, 125.93, 125.29, 124.12, 115.18, 114.90, 114.51, 70.33. Elemental analysis for C_31_H_17_N_3_OS, 1.7 H_2_O: Calculated, C—72.98, H—4.03, N—8.24, S—6.28; found, C—72.26, H—3.27, N—8.25, S—6.01. HRMS (ESI^+^): Calculated for C_31_H_17_ON_3_S (*m*/*z* 479.1092), [M + H]^+^ 480.1165; found, [M + H]^+^ 480.1164. Melting point: 234.6 °C.

*2,2′-(2-(4-(10H-Phenothiazin-10-yl)benzylidene)-1H-indene-1,3(2H)-diylidene)dimalononitrile* (**8**). In a 10-mL round flask, 4-(10H-phenothiazine-10-yl) benzaldehyde (189.6 mg, 0.625 mmol), 2,2′-(1*H*-indene-1,3(2*H*)-diylidene)dimalononitrile **5** (121.1 mg, 0.5 mmol), and a catalytic amount of piperidine were dissolved in 5 mL of acetic anhydride and heated at 38 °C for 24 h. After removing the excess solvent in a rotatory evaporator, the solid was dissolved in 100 mL of dichloromethane and washed with 60 mL of distillated water (3×). The organic phase was dried with MgSO_4_ and the solvent was removed. The crude was further purified by silica column chromatography using CH_2_Cl_2_/C_6_H_12_ (3:2) as eluent (Rf = 0.53). The grayish green solid was washed with diethyl ether to obtain 69.1 mg (yield 26%). ^1^H-NMR. (CD_2_Cl_2_, 400 MHz) δ 8.65 (d, *J* = 7.4 Hz, 1H), 8.46 (s, 1H), 8.21 (d, *J* = 9.1 Hz, 2H), 7.91 ñ 7.84 (m, 1H), 7.76 (dqd, *J* = 14.4, 7.4, 1.3 Hz, 2H), 7.54 (ddd, *J* = 7.6, 4.5, 1.2 Hz, 4H), 7.42 (td, *J* = 7.7, 1.4 Hz, 2H), 7.29 (td, *J* = 7.6, 1.3 Hz, 2H), 7.14 (d, *J* = 9.2 Hz, 2H). ^13^C-NMR. (CD_2_Cl_2_, 101 MHz, ppm) δ 144.85, 140.46, 134.82, 133.99, 133.75, 129.07, 128.01, 127.57, 126.70, 126.41, 126.30, 116.33, 113.70, 77.37. Elemental analysis for C_34_H_17_N_5_S-0.1 C_4_H_10_O-0.45 H_2_O: Calculated, C—76.07, H—3.51, N—12.89, S—5.90; found, C—75.70, H—3.14, N—12.87, S—5.55. HRMS (ESI^+^): Calculated for C_34_H_17_N_5_S (*m*/*z* 527.1205), [M + NH_4_]^+^ 545.1543; found, [M + NH_4_]^+^ 545.1545. Melting point: 287.7 °C.

## 3. Results and Discussion

Straightforward synthesis of the titled derivatives has been achieved in 3 steps starting from commercially available phenothiazine (Figure 1). Compound **1** was readily prepared with 76% yield according to reported procedures from phenothiazine and 4-bromobenzaldehyde **2** [37]. The corresponding aldehyde **1** was subsequently reacted with active methylene groups of acceptors moieties, i.e., malononitrile (**3**), 3-dicyanomethylidene-1-indanone (**4**), and 1,3-bis(dicyanomethylidene)indane (**5**), affording the push–pull chromophores **6**, **7**, and **8** with 62%, 58%, and 26% yield, respectively. The two indane-based acceptors were synthetized from 1,3-indanedione and malononitrile [38]. The conditions used for the Knoevenagel’s condensation reactions were adapted accordingly to the nature of the acceptor moiety, i.e., refluxing ethanol for malononitrile, refluxing acetic anhydride for 3-dicyanomethylidene-1-indanone, and refluxing acetic anhydride in the presence of piperidine for 1,3-bis(dicyanomethylidene) indane [39]. All synthetized chromophores were characterized by 1H, 13C NMR, high-resolution mass spectrometry, and elemental analysis.

Optical properties of the synthetized compounds **6**–**8** have been investigated by absorption spectroscopy in acetonitrile at a concentration of *ca* 1.40 × 10^−5^ M. All compounds exhibit a broad and intense band at low energy related to the internal charge transfer (ICT) taking place from the electron-rich phenothiazine unit to the electron withdrawing unit. The ICT band is red-shifted from **6** to **8** evidencing the higher electron affinity for the 1,3-bis(dicyanomethylidene)indane acceptor, which is corroborated with the electrochemical data (Table 1, Figure 1). Thus, compound **6** exhibits an ICT band centered at 410 nm (ε = 86,830 M^−1^), while for **7** the ICT is centered at 514 nm (ε = 67,470 M^−1^) and **8** at 531 nm (ε = 43,290 M^−1^).

The electrochemical behavior of the phenothiazine-based chromophores **6**–**8** has been evaluated by cyclic voltammetry in acetonitrile and the data are compiled in Table 1. The cyclic voltammograms (CV, Figure 2) of compounds **6**–**8** present one reversible oxidation peak at 0.90, 0.85, and 0.83 V, respectively, vs. Ag/AgCl associated to the formation of the radical cation on the nitrogen atom of the phenothiazine ring and two successive irreversible oxidation peaks, at similar values around 1.50 V and 1.42 V (as a shoulder) for the three compounds. Those are ascribed to the formation of the radical cation on the sulfur atom of the phenothiazine ring and the aromatic ring. In the negative potential region, an irreversible wave centered at −1.01, −0.57, and −0.39 V was observed for compounds **6**, **7**, and **8**, respectively, which is assigned to the reduction of the electron-withdrawing groups. As observed, the reduction strongly depends on the nature and strength of the acceptor. For each compound, only the first oxidation exhibits the higher potential variation due to conjugation of the nitrogen atom with the acceptor moiety. These behaviors also suggest that the phenothiazine ring is not fully conjugated with the whole π-conjugated systems and may adopt an orthogonal orientation, confirmed by standard DFT calculations (Table 2, vide infra). Furthermore, the electrochemical bandgaps (Eg^elec^) reveal similar characteristics and are in good agreement with the optical bandgaps reflecting a linear decrease in the HOMO–LUMO gap while the acceptor strength increases.

Computational DFT theoretical studies at the DFT level of the chromophores **6**–**8** were performed using the (B3LYP/6-31G(d,p)) basis set (Table 2) [40]. All HOMO levels are located along the phenothiazine ring system, whereas the LUMO levels are mainly located on the acceptor and the π-conjugated spacer. The energy values of the calculated frontier orbitals can be correlated with the cyclic voltammetry experiments for the HOMO levels while the LUMO levels appear at higher energies than expected. These main differences could be explained by the fact that in the excited state, the compounds exhibit a higher polarization due to the ICT, which is more stabilized by solvation effects in CH_3_CN giving lowered LUMO energy levels.

The anion binding properties of the compounds **6**–**8** were investigated by UV–visible spectroscopy with several anions such as PF_6_^−^, HSO_4_^−^, Cl^−^, Br^−^ NO_2_^−^, CH_3_CO_2_^−^, and CN^−^ as tetrabutylammonium salts in CH_3_CN. Even in polar solvent, these anionic guests interact with the chemosensors **6**–**8** leading to different spectroscopic changes that are structure-dependent and more specifically to the nature of the acceptor unit. Spectroscopic changes are associated to batho-, hyper-, or hypochromic effects of the ICT band. The optical changes and their amplitude as well as the presence of isosbestic points are ascribed to the ability of chemosensors to interact more or less specifically with a given anion.

The three push–pull phenothiazine derivatives exhibit different behaviors, as shown in Figure 3, Figure 4 and Figure 5. Hence, compounds **7** and **8** displayed the most significant spectroscopic changes while compound **6** did not undergo any changes regardless of the anion tested (Figure 3). In addition, net disparities emerged between **7** and **8** (Figure 4 and Figure 5) depending on the nature of the analyte. In both cases, trivial or no interaction are seen for PF_6_^−^, HSO_4_^−^, Cl^−^, and Br^−^, whereas with NO_2_^−^, CH_3_CO_2_^−^, and CN^−^, the main spectral modifications are attained. Moreover, chromophore **7** interacts only the cyanide anion while **8** gives a colorimetric response with NO_2_^−^, CH_3_CO_2_^−^, and CN^−^.

Thus, optical properties of compound **6** remain unchanged regardless of the tested anions, even with an excess. On the contrary, the addition of 1 equivalent of cyanide anion to chromophore **7** induces the formation of the adduct product 2-(2-((4-(10*H*-phenothiazin-10-yl)phenyl)(cyano)methyl)-3-oxo-2,3-dihydro-1*H*-inden-1-ylidene)malononitrile. **7.CN^−^** (Figure 5) associated with a striking hypochromic effect and a hypsochromic shift, highlighting the nucleophilic conjugate addition of the cyanide on the β-vinylic carbon of the p-conjugated system as depicted in Figure 6, and altering the push–pull effect in agreement with previously reported results on similar systems [33].

In the case of chromophore **8**, the recognition mode appears to be different as a bathochromic shift of 561 nm, 571 nm, and 576 nm is observed for NO_2_^−^, CN^−^, and CH_3_CO_2_^−^ anions, respectively. In addition to the red shift of the maximum of absorption, a fine vibronic structure appears for the ICT band. This behavior is in favor of the presence of anion–π interactions related to the π-acceptor strength [41,42,43] instead of nucleophilic conjugate addition, as seen for chromophore **7** (Figure 7).

Interestingly, the strongest bathochromic shift is observed for the unexpected CH_3_CO_2_^−^ anion. This result suggests a greater affinity of **8** towards the acetate anion compared to the others even if its pK_b_ (9.25) is two-fold higher than the pK_b_ (4.60) of cyanide. To confirm this hypothesis, competitive titration assays were conducted with **8.CH_3_CO_2_^−^** by the concomitant addition of 1 equivalent of the most competitive NO_2_^−^ and CN^−^ anions (Figure 8). The absorption spectrum of **8.CH_3_CO_2_^−^** remained unchanged, testifying to the stronger specificity of **8** for the acetate anion. Similar results were obtained when CH_3_CO_2_^−^ was added to a solution of the complex **8.CN**^−^ or of **8.NO_2_^−^** (Figure 9). In addition, experiments conducted between the complex **8.CN^−^** and NO_2_^−^ demonstrate the higher ability of **8** to interact with the cyanide anion, corroborating the observed spectroscopic changes and following the pK_b_ range (pK_b_ = 10.85 for nitrite anion). For the other anions having a pK_b_ higher than ~11 (pK_b_ (HSO_4_^−^) = 17, pK_b_ (Cl^−^) = 20, pK_b_ (Br^−^) = 22, pK_b_ (PF_6_^−^) > 22 [44]), no interaction is observed. Nevertheless, not only must the pK_b_ be taken into consideration to explain the trend, but also the symmetry, geometry, and size of the anions. On the basis of these results, a series of affinity can be settled as follows for chromophore **8**: CH_3_CO_2_^−^ > CN^−^ > NO_2_^−^ >>> Cl^−^ ~ Br^−^ ~ HSO_4_^−^ ~ PF_6_^−^.

The selectivity of chromophore **7** was also investigated by UV–vis spectroscopy (Figure 10). Upon the addition of putative competitive anions, the spectrum of **7.CN^−^** remained unchanged, highlighting the fact that the addition of the cyanide is irreversible and unaffected by the presence of other ions in the medium. This indicates that the chemosensors are selective and specific to cyanide over the other anions. The same behaviors are attained when chromophore **7** is mixed with 1 eq. of CN^−^ and 1 eq. of each anion.

Job-plots were performed in CH_3_CN and indicate, for the studied anions, the formation of either a complex [1:1] for **8.CN^−^** and **8.NO_2_^−^** (as exemplified in Figure 11) or a complex [1:2] for **8.CH_3_CO_2_^−^**. 

Association constants K_i_ were determined by solving the nonlinear equations given in literature [45]. For instance, for complex [1:1], [H_0_] = [H] + [G], [G_0_] = [G] + [HG] and K_1_[H].[G] = [HG]; for complex [1:2], [H_0_] = [H] + [G] + [HG], [G_0_] = [G] + [HG] + [HG_2_], K_1_[H].[G] = [HG], and K_2_[HG].[H] = [HG_2_] (where H corresponds to the host, G the guest, and the subscript 0 denotes the initial concentration for each species). The total absorbance is given by the equation ${A=\varepsilon }_{H}{[H]+\varepsilon }_{\mathrm{HG}}{[\mathrm{HG}]+\varepsilon }_{{\mathrm{HG}}_{2}}{[\mathrm{HG}}_{2}{]+\varepsilon }_{G}\left[G\right]$, where ε corresponds to the molar extinction coefficient for each species. Hence, the determination of the association constants K_1_ gives the following trend *K*^CH_3_CO_2_−^ >> *K*^CN−^ > *K*^NO_2_−^ (Table 3), which is in good agreement with the experimental optical changes observed in UV–Vis spectra. In addition, based on the equilibrium constants and ε, we are able to predict a reasonable titration theoretical spectrum (Figure 12) matching with the experimental data.

According to the calculations, only 7.2% of CH_3_CO_2_^−^ remains free in solution vs. 26.8% and 36% for CN^−^ and NO_2_^−^, respectively. Both experimental and calculation findings pinpoint that **8** interacts more strongly with CH_3_CO_2_^−^ than CN^−^ and NO_2_^−^.

Finally, as proof of concept for practical applications, a paper test strip was prepared as follows: a white paper strip was dipped into a 10^−3^ M solution of **7** in CH_3_CN for 1 min, then air-dried. After this process, the paper took on a red wine color tint. The immersion of the latter into a solution of CN^−^ anions colorized the test strip in orange, whereas for the others (PF_6_^−^, HSO_4_^−^, Cl^−^, Br^−^ NO_2_^−^, CH_3_CO_2_^−^), the strip’s color remained unchanged. The same behavior was observed when the paper test strip was immersed in a solution containing all anions together (Figure 13). These results clearly demonstrate the possibility of using this strip as qualitatively naked eye sensing of cyanide in complex solutions. The method is very cheap and does not require any calibration for qualitative detection.

## 4. Conclusions

In summary, we reported herein the straightforward synthesis and characterization of novel phenothiazine derivatives possessing remarkable sensitivity and selectivity for cyanide (LOD = 9.12 µM) over other anions. The sensing mechanism is based on the irreversible nucleophilic addition of the cyanide leading to an irreversible change of the color. This behavior makes it a system of choice for naked eyes sensing devices (Table 4) based on the absorption properties since most of the phenothiazine-based chemosensors developed up to now have been used as fluorescent probes. Interestingly, by changing the strength of the acceptor, the mechanism is switched into an anion-p interactions recognition mechanism associated to almost a two-fold increase in the LOD for cyanide detection reaching a value of 4.59 µM. In addition, the enhancement of the acceptor strength has allowed the possibility to selectively detect nitrite and acetate. Experimental data demonstrated that the highest binding ability is achieved for the acetate anion even in presence of nitrite and cyanide with a LOD of 2.68 µM. A paper test strip was efficiently prepared and used to sense qualitatively cyanide anions in solutions. Further studies are in progress striving for new chemosensors suitable for sensing applications as well as their implementation in electronic devices such as field effect transistors [46].

## Data Availability

The data presented in this study are available on request from the corresponding author.

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
