# Peer review of "Naked-Eye Chromogenic Test Strip for Cyanide Sensing Based on Novel Phenothiazine Push–Pull Derivatives"

_biosensors, 2022, doi:10.3390/bios12060407_

Round 1
Reviewer 1 Report
In the present paper, the authors are describing 3 new phenothiazine derivatives forming molecular probes for detecting cyanide using the strong ICT absorption band in the visible region. The paper is well presented, and the compounds prepared are characterized according to standard methods. The discussion of results is properly done showing the scope of the probes in sensing of cyanide in different conditions. The comparison with other methods is provided in a good shape. The only missing point in such investigation is about the presence or not of the emission properties of such new compound which could give to the paper and readers interesting results. The paper may be publish in the present form.
Author Response
Response to reviewer 1 comments :
We kindly thank the reviewer for his comment.
Regarding his remark on fluorescence, we have conducted fluorescent measurements but the target compounds don't exhibit fluorescent properties as already mentioned in the original paper (ligne 69) as follow :
"In addition, the functionalization of the phenothiazine pattern at the nitrogen atom vanished its intrinsic fluorescence properties, making the newly synthetized compounds as only colorimetric sensors".
Reviewer 2 Report
Authors should research and discuss the effects of some common anions, such as carbonate ions, phosphate salts, and sulfide ions, etc, on the sensing properties of compounds 6, 7 and 8 for the detection of cyanide and acetate anion, respectively, as possible.
Author Response
Response to Reviewer 2 comments.
We thank the reviewer for his remarks and comments.
It would be certainly an added value and interesting feature to analyse regarding such mentioned anions. Unfortunately the former student who performed the present work is back to his home country during the pandemic crisis and we will be unable to conducted the asked comparison analysis.
Reviewer 3 Report
The authors developed a new set of sensors based on phenothiazine derivatives for the optical detection of cyanide. For the three investigated compounds authors measured the electrochemical properties and UV-Vis spectra changes upon anion binding. One of the compounds showed a high selectivity for cyanide ions and was selected as a prospective tool for naked-eye cyanide sensing.
The manuscript is easy to read and the described results are clear. Experimental methods are described in detail.
Minor revision: A few details on the methods applied in quantum chemical calculations and the method of calculating the theoretical spectrum in Figure 12 may be useful for the general reader and can be added in the Methods section.
Author Response
Response to Reviewer 3 comments
We firstly thank the reviewer for his valuable comment.
As suggested, we have added in the Method section a brief description and details about the quantum calculations and the method of calculating the theoretical spectrum in Figure 12.
Round 2
Reviewer 2 Report
The manuscript reports that synthesis of novel phenothiazine-based push-pull (D-π-A) chemosensors, which will be optically sensitive to the nucleophilic addition of the cyanide suitable for naked eye cyanide sensing. The results are very interesting, especially, a paper test strip was efficiently prepared and used to sense qualitatively cyanide anions in solutions. The paper should be published in the Journal.